# Loss of the yeast transporter Agp2 upregulates the pleiotropic drug-resistant pump Pdr5 and confers resistance to the protein synthesis inhibitor cycloheximide

Yusra Manzoor[1]☉, Mustapha Aouida[1]☉, Ramya Ramadoss[2], Balasubramanian Moovarkumudalvan[2,3], Nisar Ahmed[3], Abdallah Alhaj Sulaiman[1], Ashima Mohanty[3], Reem Ali[1], Borbala Mifsud[3], Dindial Ramotar[1]*

1 Division of Biological and Biomedical Sciences, College of Health and Life Sciences, Hamad Bin Khalifa University, Education City, Qatar Foundation, Doha, Qatar, 2 Mahatma Gandhi Medical Advanced Research Institute (MGMARI), Sri Balaji Vidyapeeth (Deemed to be University), Puducherry, India, 3 Division of Genomics and Precision Medicine, College of Health and Life Sciences, Hamad Bin Khalifa University, Education City, Qatar Foundation, Doha, Qatar

☉ These authors contributed equally to this work.
* dramotar@hbku.edu.qa

## Abstract

The transmembrane protein Agp2, initially shown as a transporter of L-carnitine, mediates the high-affinity transport of polyamines and the anticancer drug bleomycin-A5. Cells lacking Agp2 are hyper-resistant to polyamine and bleomycin-A5. In these earlier studies, we showed that the protein synthesis inhibitor cycloheximide blocked the uptake of bleomycin-A5 into the cells suggesting that the drug uptake system may require *de novo* synthesis. However, our recent findings demonstrated that cycloheximide, instead, induced rapid degradation of Agp2, and in the absence of Agp2 cells are resistant to cycloheximide. These observations raised the possibility that the degradation of Agp2 may allow the cell to alter its drug resistance network to combat the toxic effects of cycloheximide. In this study, we show that membrane extracts from *agp2Δ* mutants accentuated several proteins that were differentially expressed in comparison to the parent. Mass spectrometry analysis of the membrane extracts uncovered the pleiotropic drug efflux pump, Pdr5, involved in the efflux of cycloheximide, as a key protein upregulated in the *agp2Δ* mutant. Moreover, a global gene expression analysis revealed that 322 genes were differentially affected in the *agp2Δ* mutant *versus* the parent, including the prominent *PDR5* gene and genes required for mitochondrial function. We further show that Agp2 is associated with the upstream region of the *PDR5* gene, leading to the hypothesis that cycloheximide resistance displayed by the *agp2Δ* mutant is due to the derepression of the *PDR5* gene.

## Introduction

Agp2 was first shown to be an uptake transporter of L-carnitine, which is intracellularly required for the complete oxidation of acetyl-CoA derived from fatty acid β-oxidation [1].

**Data Availability Statement:** All relevant data are within the manuscript and its Supporting Information files.

**Funding:** Funds from Qatar Foundation provided to the College of Health and Life Sciences. The funders had no role in study design, data collection and analysis, decision to publish, or preparation of the manuscript

**Competing interests:** he authors have declared that no competing interests exist.

Agp2 belongs to a family of low-affinity and non-specific amino acid permeases such as Put4, Alp1, Lyp1, Can1, and Gap1 which are operative when the nutrients are limited [2–4]. Subcellular fractionation and immunoelectron microscopy studies revealed that the Agp2 protein has multiple localizations within the cell [1]. These include the plasma membrane and membranes derived from the peroxisome, the endoplasmic reticulum, and the mitochondria, suggesting that Agp2 plays an essential role in the distribution of substrates in the cell [1]. In 2004, we rediscovered the *AGP2* gene from a genome-wide screen of a yeast haploid mutant collection that when deleted confers increased resistance to bleomycin-A5, an analog with a spermidine moiety of the bleomycin anticancer drug family, which damages the DNA [5]. Mutants devoid of Agp2 are extremely resistant to bleomycin-A5 and very weakly accumulate fluorescently labeled bleomycin-A5 (F-BLM) into the cells demonstrating that the Agp2 protein plays a major role in the uptake of the drug [5]. Of note, no other amino acid permeases such as the general amino acid permease Gap1 and the functionally related L-carnitine transporter Hnm1, were identified from the genome-wide screen as gene-deletion mutants displaying resistance to bleomycin-A5 [5], raising the possibility that Agp2 plays a specific role in the uptake of bleomycin-A5 into the cells.

Besides L-carnitine and bleomycin-A5, Agp2 can mediate the high-affinity uptake of spermine and spermidine as cells lacking Agp2 (*agp2Δ* mutant) displayed sharply reduced initial velocity in the uptake of labeled polyamines [1, 6]. This defect of the *agp2Δ* mutant overlapped with its hyper-resistant phenotype towards the toxic effects of polyamines [6]. In 2007, Uemura *et al*., reported the isolation and characterization of two new high-affinity polyamine uptake permeases, Dur3 and Sam3, that account for the greater fraction of polyamine uptake in the yeast cells [7]. However, *dur3Δ sam3Δ* double mutants deleted for both the *DUR3* and *SAM3* genes are not as resistant to the toxic effects of polyamines as compared to the *agp2Δ* single mutant, suggesting that Agp2 is a dominant factor conferring resistance to polyamines [8].

Several observations challenged the notion of whether Agp2 is a high-affinity transporter or merely a sensor that influences the activity level of specific transporters dedicated to either the uptake or efflux of toxic compounds. *S. cerevisiae* possesses many sensors or regulators serving as non-transporting transceptors that are embedded in the plasma membrane [8, 9]. One such sensor is Ssy1, which belongs to the same amino acid permease family as Agp2, but it cannot function as a transporter [10–12]. Ssy1 senses amino acid levels and promotes the expression of the amino acid permeases including *AGP1*, *BAP2*, *BAP3*, *DIP5*, and *TAT1*. Sensing by Ssy1 triggers the proteolytic activation of the transcription factors, Stp1 and Stp2, which translocate from the cytosol to the nucleus to stimulate the expression of the amino acid permease genes [10, 13, 14]. In addition to Ssy1, there is a broader list of plasma membrane transceptors that serve as sensors including the ammonium and general amino acid permeases Mep2 and Gap1, respectively [11, 12]. Agp2 possesses the functional characteristics of sensors, in particular, regulating the expression of plasma membrane transporter genes such as *DUR3*, *SAM3*, *HNM1*, *TPO2*, and *HXT3*, and may perform its role in sensing and controlling the uptake of diverse substrates [1, 6–8, 15–22].

The notion that Agp2 could be a sensor or regulator seems more likely based on the following observations: (i) The *agp2Δ* mutant showed reduced transcription of the genes encoding the polyamine transporters Dur3 and Sam3, as well as the L-carnitine transporter Hnm1, suggesting that Agp2 plays a role in regulating gene expression similar to the non-transporting transceptor Ssy1 [10–12], and (ii) The *agp2Δ* mutant is strikingly resistant to other aromatic and cation molecules such as the anticancer drug anthracyclines and the protein synthesis inhibitor cycloheximide (CHX) [5, 21, 23]. We focused our attention on the effect of CHX on Agp2, as our previous study showed that low concentrations of this drug can terminate the uptake of F-BLM into the cells [20]. While this data can be explained if CHX directly competes

or inhibits F-BLM uptake, our recent study by Mohanty et al., 2023 showed that *agp2Δ* mutants are hyper-resistant to CHX [23]. Moreover, CHX can rapidly trigger the disappearance of Agp2-GFP, in a concentration- and time-dependent manner [23]. The same study also showed that CHX induces the ubiquitinylation of Agp2-GFP to higher molecular weight forms that disappeared within 10 minutes following the drug exposure [23]. These observations strongly suggest that Agp2 is undergoing degradation upon encountering the toxic compound CHX and this may send a signal to diminish the level of the drug in the cell. It is not entirely clear how CHX enters the cells, and non-radioactively labeled CHX is not commercially available to monitor the kinetics of its uptake. However, the Agp2-controlled broad-substrate specificity uptake transporters Dur3 and Sam3 are involved in the process. Cells lacking either Dur3 or Sam3 showed parental sensitivity to CHX, while deletion of both *DUR3* and *SAM3* caused the resulting double mutant *dur3Δ sam3Δ* to exhibit resistance to the drug. Interestingly, the *agp2Δ* mutant is significantly more resistant to CHX than the *dur3Δ sam3Δ* double mutant [23], suggesting that Agp2 controls another mechanism besides the uptake transporters Dur3 and Sam3 that is required to confer resistance to the toxic effects of CHX. Since Agp2 is present on the membranes of multiple organelles and can regulate the transcription of several membrane transporter genes, we set out to analyze in total membrane extracts whether there are proteins affected by the absence of Agp2 that can account for the CHX resistance phenotype.

## Materials and methods

### Strains, media, transformation, plasmid, vector, and reagents

The *S. cerevisiae* strains used in the present study are BY4741 wild type (WT) with the genotype *MATα his3Δ leu2Δ met15Δ ura3Δ* and the isogenic gene deletion mutants *agp2Δ*::KAN and *pdr5Δ*::KAN. Yeast cells were grown at 30°C in either Yeast Peptone Dextrose (YPD, FOR-MEDIUM CCM0105) YPD [1% (w/v) or minimal synthetic (SD: 0.65% yeast nitrogen base without amino acids, 2% dextrose, 0.17% dropout mix) medium used for transformation [24–26]. All strains used were obtained from non-essential haploid mutant resources from this laboratory. All chemical reagents including CHX were purchased from Sigma, St Louis, USA. The plasmid expressing Agp2-GFP and the vector carrying GFP were previously described [5].

### Spot test and cell growth analyses

Standard spot tests were performed as previously described [27]. For the growth curve, the cells were cultured overnight in YPD media then the OD $600_{nm}$ was adjusted to ~ 0.2 with fresh YPD media in a 96-well plate. Drugs were added to the cells at the indicated concentrations and growth was monitored for 24 hours in a TECAN plate reader with shaking. The temperature was set to 30°C and the OD was taken every two hours. The result was plotted as a graph of OD $600_{nm}$ against the time [28].

### Preparation of total membrane extract

Total membrane fractions were prepared as described by Drew et al., [29]. Briefly, yeast cells were cultured overnight at 30°C with shaking, and log-phase cells were pelleted and re-suspended in yeast suspension buffer (YSB) (50 mM Tris–HCl (pH 7.6), 5 mM EDTA, 10% glycerol, 1× complete protease inhibitor cocktail tablet) along with sterile glass beads (0.5 mm diameter, BioSpec Cat. No. 11079105). Yeast cells were lysed using a bead mill homogenizer (BeadMill 4, FisherScientific) at 5 m/s for 5s and repeated 10 times with cooling on ice in between. The unbroken cells and glass beads were spun down by brief centrifugation at 3000

rpm for 1 min in an Eppendorf microcentrifuge. The supernatant containing total cell extracts (aliquots saved as total cell extract) was further centrifuged at 4°C for 1 hour at 15,000 rpm (20,000 x g) in a benchtop microcentrifuge to obtain the total membrane fraction as a pellet. The pellet was resuspended in 50 μl YSB buffer. The total membrane fraction was quantified before processing for SDS-PAGE, Western blot, and immunoprecipitation analyses. The experiment was repeated independently at least three or four times.

## Mass spectrometry analysis of the total membrane fractions from the WT and the *agp2Δ* mutant untreated and treated with CHX

The protein samples from each of the four groups indicated in the text were analyzed with three biological replicates. The protein was in-gel reduced and alkylated, then digested with the addition of Promega sequencing grade modified trypsin and incubated overnight at 37°C. The resulting peptides were extracted before drying and resuspending in aqueous 0.1% TFA for LC-MS. LC-MS/MS was performed with elution from a 50 cm C18 EasyNano PepMap column over 1 h driven by a Waters mClass UPLC onto an Orbitrap Fusion Tribrid mass spectrometer operated in DDA TopSpeed mode with a 1 s cycle time. MS1 spectra were acquired in the Orbitrap mass analyzer at 120K resolution and MS2 spectra were acquired in parallel in the linear ion trap following HCD fragmentation. The resulting LC-MS chromatograms in Thermo.raw format were imported into Progenesis QI for peak picking and alignments. A concatenated MS2 peak list in.mgf format was exported and searched using the Mascot search program against the *Saccharomyces cerevisiae* subset of the SwissProt proteome, appended with common proteomic contaminants. Matched peptides were filtered using the Percolator algorithm to achieve a 1% peptide spectral match false discovery rate, as assessed empirically against a reverse database search. Peptide identifications were imported onto Progenesis QI-aligned LC-MS chromatograms and matched between acquisitions. Identified MS1 peak areas were integrated and compared for relative peptide quantification of non-conflicting peptide sequences. Relative protein quantification was inferred from underlying peptide values following normalization to total peptide intensity. Final accepted protein quantifications were filtered to require a minimum of two quantified peptides. Statistical testing was performed using ANOVA with p-values converted to q-values for multiple test correction using the Hochberg and Benjamini approach. Values less than 0.05 were considered significant hits. Further, the data thus generated was analyzed using the DEP R package [30].

## RNA extraction and reverse transcription (RT)-PCR analysis

Total RNA was extracted from the parent BY4741 and the *agp2Δ* mutant yeast strains that were grown overnight in 10 ml of YPD liquid media using the RiboPure-Yeast extraction kit (Ambion) and DNA-free kit (Ambion) to remove genomic DNA contamination. The total RNA extracted was quantified using a spectrophotometer at $A_{260}$ wavelength, and 2 μg of total RNA was then converted to cDNA using a High-Capacity cDNA Reverse Transcription Kit as per manufacturer protocol (Applied Biosystems). The PCR primers used for the RT-PCR for the gene of interest were PDR5-F, 5′ CGTTACTAGCTACTCCTCCGCGTCT–3′; PDR5-R, 5′ TGGGTTTAGGCAACC ATCCATTGGTTTCAA–3′ (generating a product of 520 bp), *ACT1* gene was used as a control and the primers used were ACT1-F, 5′–TGGGTATCCAAGCAC ATCAA–3′; ACT1-R, 5′–TGATAAACCCGCTGAA CACA–3′. The PCR reaction was carried out using AmpliTaq Gold 360 Master Mix and the PCR program used was as per manual. Briefly, the reaction was set for 3 min at 95°C followed by 25 cycles of 30 seconds of denaturation at 95°C, 30 seconds of annealing at 55°C, and 1 minute and 30 seconds for primer

extension at 72˚C. The final extension was carried out for 10 min at 72˚C. The PCR products were run on 1% agarose gel.

### RNA-seq data analysis

RNA-sequencing (RNA-seq) of total RNA was conducted at the WCMQ Genomics Core Facility (Doha, Qatar). The *agp2Δ* mutant and the WT strain were cultured in YPD liquid media without treatment and the total RNA was extracted from three independent experiments and subjected to bulk RNAseq analysis to quantitate cellular RNA concentrations. Briefly, 400 ng of high-integrity total RNA was used to generate strand-specific 300–400 bp libraries using NEXTflex Rapid Directional RNA-Seq Kit (Bioo-Scientific, USA Catalog #NOVA-5138-07) according to the manufacturer's protocol. Library quality and quantity were analyzed with the Bioanalyzer 2100 (Agilent, USA) on a high-sensitivity DNA chip. The libraries were then pooled in equimolar ratios and paired-end 150bp sequenced on Illumina Next-Seq 550.

The raw RNA-seq reads were checked for quality control (QC) using FastQC [31] and trimmed with TrimGalore. The trimmed reads were aligned to the *Saccharomyces cerevisiae* genome downloaded from the Ensembl genome browser with STAR [32]. RSEM was used to calculate the expression values as expected count from the aligned RNA-seq data [33]. For further analysis, RSEM-STAR output was imported into R (version 4.2.0, https://www.r-project.org). Differentially expressed genes were extracted using DESeq2 [34]. For visualization by volcano plots, the $log_2$foldchange (l2fc) value from DESeq2 was shrunk using the apeglm algorithm [35]. Genes were regarded as being differentially expressed when meeting the criteria of absolute 2.5 l2fc and $p$-value $< 0.05$. The differentially expressed genes in mutant versus wild type were submitted to gene ontology analysis using the Profiler web server. Heatmaps were generated using the R package pheatmap [36]. Generation of tables and plots was performed using the R packages dplyr [37], and ggplot2 [38].

### Accession numbers

For the project named LC-MS/MS quantification of proteome derived from the WT and the *agp2Δ* mutant strains of Yeast, the accession number is PXD037674, and the hyperlink to the data is https://www.ebi.ac.uk/pride/archive/projects/PXD037674. For the RNAseq data, the accession number is CRA008668 and the hyperlink to the data is https://bigd.big.ac.cn/gsa/browse/CRA008668.

### Protocol for ChIP for yeast cells

ChIP for BY4741-GFP and BY4741-Agp2-GFP was carried out as previously described [39], but with the following modifications, as follows: (i) **Growth and cross-linking of protein and DNA complex**—BY4741-GFP (control strain) and BY4741-Agp2-GFP were grown overnight at 30˚C in 20 ml of Sc-Ura media until the OD reached 20. The proteins were cross-linked onto the DNA by adding 550 µl of 37% formaldehyde (1% final concentration) and incubated at room temperature for 20 mins on a rotator. To stop the cross-linking reaction, 3 ml of heat-sterilized 2.5M glycine was added and incubated at room temperature for 5 mins on a rotator. The cells were then harvested by centrifugation at 2500 x g for 5 mins at 4˚C. The supernatant was discarded, and the pellet was resuspended in 5 ml of ice-cold TBS. This step was repeated twice. After the final centrifugation, the pellet was resuspended in 1 ml of ice-cold FA-lysis buffer (50 Mm Hepes pH 7.5, 150 mM NaCl, 1 mM EDTA, 1% Triton-X-100, 0.1% sodium deoxycholate and 0.1% SDS) and centrifuged again at 3,000 rpm for 5 mins at 4˚C. (ii) **Cell lysis and DNA shearing**—The cells were lysed by resuspending the pellet in 1 ml of ice-cold

FA lysis buffer/2mM PMSF and the sample was transferred to a 2 ml microcentrifuge tube containing silica-zirconia beads. The cells were lysed using a mini bead beater at maximum speed for 3 minutes. The samples were placed on ice for 1 minute to avoid overheating of the sample. These steps were repeated 5 times for a total breakage time of 18 minutes. The supernatant was then carefully transferred to a new tube and centrifuged at maximum speed for 15 minutes at 4˚C. The supernatant was discarded, and the pellet was resuspended in 500 μl of ice-cold FA lysis buffer. The sample was then sonicated to shear the DNA for 30 seconds using a continuous pulse at a power output of 20%. The sample was placed on ice for 1 minute to avoid overheating. These steps were repeated twice. The sample was then centrifuged at maximum speed for 30 minutes at 4˚C and the supernatant was then transferred to a new tube containing 1 ml ice-cold FA lysis buffer, and (iii) **Immunoprecipitation**—To immunoprecipitate DNA bound to GFP or Agp2-GFP, 350 μl of the sample was incubated with 7 μl of anti-GFP antibody (Roche, Cat # 11814460001) overnight at 4˚C. The next day, 25 μl of magnetic protein G beads were washed 3 times with 500 μl of ChIP wash buffer (10 mM Tris, 1 mM EDTA, 0.5% NP40, and 0.5% sodium deoxycholate). The magnetic beads were then incubated with the sample containing anti-GFP antibody for 1 hr at 4˚C, and then for 1 hour at room temperature. The magnetic beads were pulled down using a magnetic stand and the supernatant was removed and added to a new tube. This is the input sample. The beads were then washed 3 times with 500 μl ChIP wash buffer. After the final wash, 30 μl of TE buffer and 7 μl of Proteinase K were added to the beads, and the sample was placed in a PCR machine to elute and reverse cross-link the protein. The sample was incubated for 2 hours at 42˚C, followed by 6 hours at 65˚C and 10 minutes at 95˚C. The DNA was then extracted by using the phenol-chloroform extraction method and Real-Time PCR was carried out using PowerUp SYBR Green Master Mix according to the manufacturer's protocol.

## Results

### The *agp2Δ* mutant confers resistance to CHX

To examine whether the *agp2Δ* mutant is indeed resistant to the toxic effects of the protein synthesis inhibitor CHX, as compared to the parent, we conducted two independent assays. In the first assay, overnight cultures were adjusted to O.D 600 of ~ 0.6, serially diluted, and spotted onto solid YPD agar plates with the indicated concentrations of CHX (Fig 1A).

A dilution of 1:10 of the *agp2Δ* mutant grew on the solid media with 0.71 μM of CHX, but not the WT (Fig 1A). In the second independent assay, the overnight cultures were adjusted to a low initial O.D 600 of ~ 0.15 in fresh liquid YPD without and with 0.14 μM CHX, and the OD readings of the cultures were taken every 2 hr to monitor the growth of the strains over a 22-hr period (Fig 1B). The *agp2Δ* mutant grew in the presence of CHX to reach an OD of ~ 1.8 in 22 hr, while the WT under the same condition reached an OD of only ~ 0.2 (Fig 1B). Both assays showed that the *agp2Δ* mutant is resistant to CHX.

### Analysis of the total membrane extracts reveals proteins with differences in expression level between the WT and the *agp2Δ* mutant

Since Agp2 can respond rapidly to CHX [23], we examined whether there could be differences amongst the membrane proteins between the WT and *agp2Δ* mutant following acute treatment with CHX. Briefly, exponentially growing cultures in liquid YPD were treated without and with CHX (3.5 μM CHX for 15 mins) and the total membrane fractions were prepared from three independent experiments and analyzed by one-dimensional gel stained with coomassie (Fig 2).

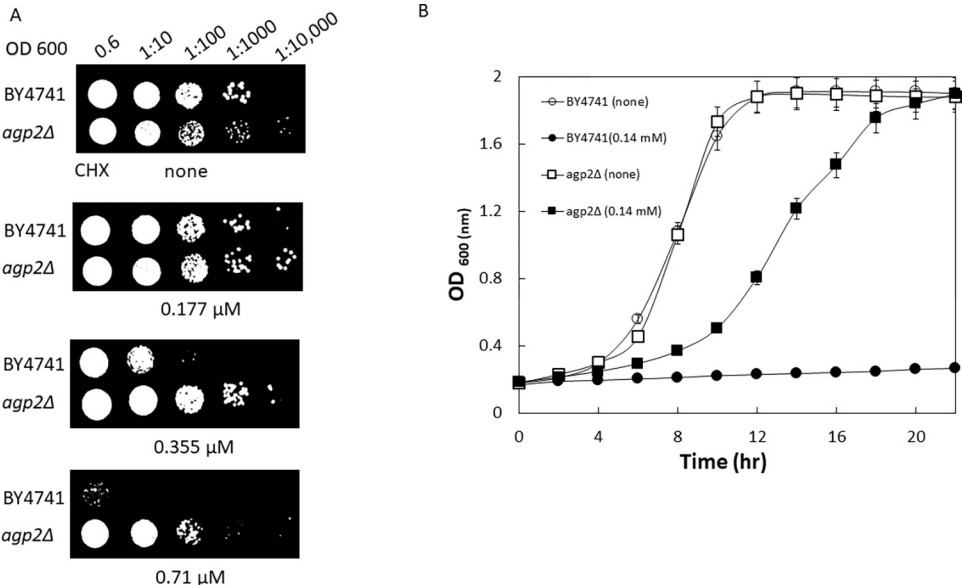

**Fig 1. The *agp2Δ* mutant confers resistance to CHX. A**, spot test analysis to determine CHX sensitivity of yeast strains. BY4741 and *agp2Δ* strains of OD $_{600}$ ~0.6 were serially diluted and spotted onto YPD agar containing increasing concentrations of CHX 0, 0.177, 0.355, 0.71 and 1.77 μM. The plates were grown at 30˚C and imaged after 48 hrs. **B**, Growth rate of cells in the absence and presence of CHX. The results are representative of three independent experiments.

The analysis revealed that several proteins were differentially expressed between the WT and the *agp2Δ* mutant (Fig 2). For example, an ~ 52 kDa polypeptide was more abundant in the WT as compared to the *agp2Δ* mutant (Fig 2 lane 1 vs. 2, purple arrow), and this same ~ 52 kDa polypeptide was induced by CHX treatment in the WT, but not in the *agp2Δ* mutant (Fig 2, lane 3 vs. 4, purple arrow). Likewise, the blue, green, and brown arrows indicated polypeptides that were more abundant in the WT than the *agp2Δ* mutant, whereasthe red arrow indicated a polypeptide that was more abundant in the *agp2Δ* mutant as compared to the WT (Fig 2, see legend), suggesting that cells devoid of Agp2 may have defects in maintaining the cellular level of specific proteins.

## Mass spectrometry reveals that the total membrane fractions derived from the WT and the *agp2Δ* mutant contain differentially expressed proteins

To confirm the above observations, we subjected the total membrane extract to LC-MS/MS analysis to identify the proteins with unique differences (see Materials and Methods). The membrane samples comprised four groups, WT untreated (-), WT treated with CHX (+), *agp2Δ* untreated (-), and *agp2Δ* treated with CHX (+). Of the total proteins identified by mass spectrometry, 1,122 were accepted for relative quantification between the samples, among which 46 proteins (shaded in green) were determined to be significantly different between the four groups (S1 Table, see column H&B q-value). Among the 1,122 proteins, 34 were represented by five or more unique peptides, criteria set by Gilbert et al 2005 representing highly positive hits [40], and these showed a maximum fold difference of ≥ 4.0 (S2 Table). Interestingly, of the 34 proteins, four (e.g., Pdr5 and Pdr15) are involved in the multidrug resistance pathway and 17 perform various roles in the mitochondria. A principal component analysis (PCA) plot (S1 Fig) of the top 500 expressed proteins was generated to obtain a broader overview of the data and to observe the contrasts among the groups. In general, a significant

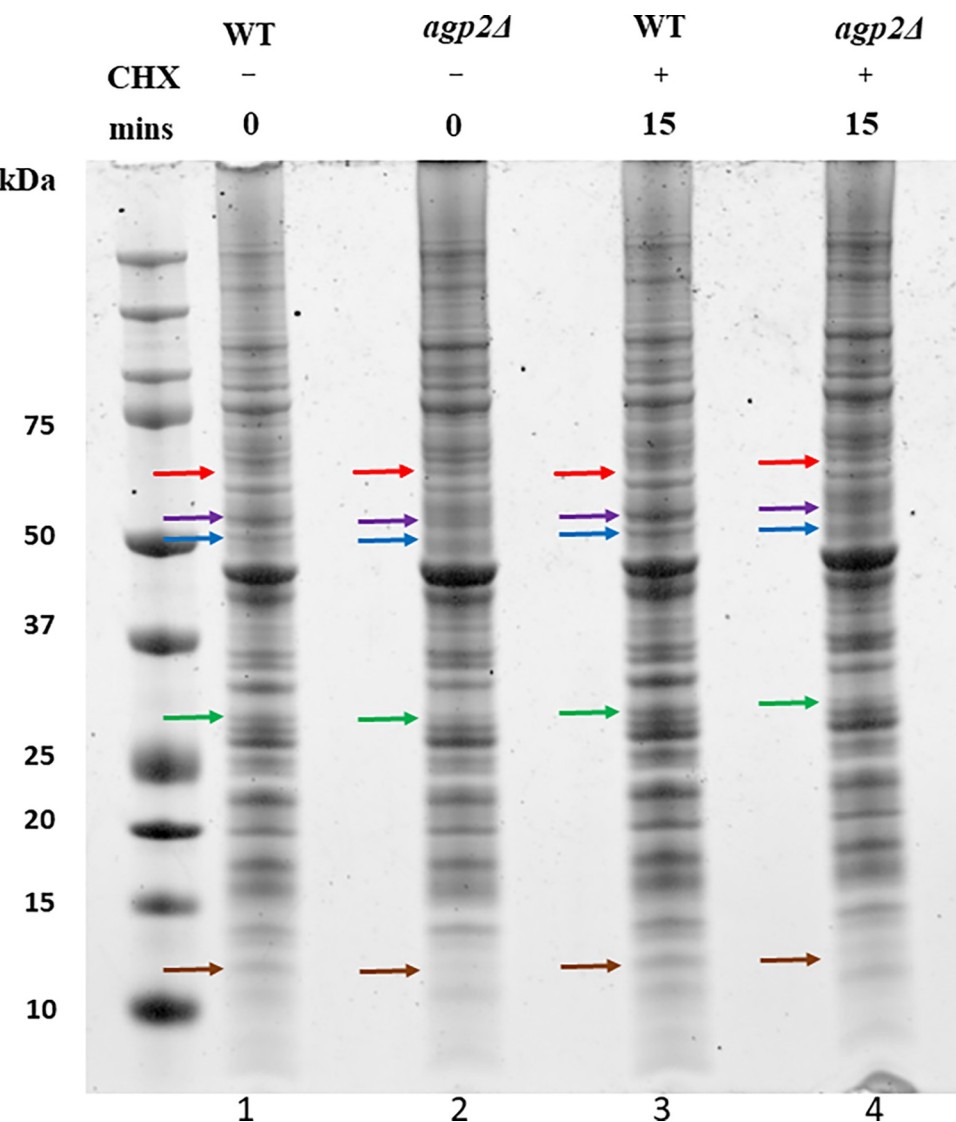

**Fig 2. One-dimensional PAGE analysis of crude plasma membrane extracts showing proteins with differences in expression level between the WT and the *agp2Δ* mutant.** One-dimensional 8% SDS-PAGE gel analysis of crude plasma membrane extract from the BY4741 (WT) and the *agp2Δ* strains. Lane 1 WT and lane 2 *agp2Δ* were extracts prepared from cells without CHX treatment, and lane 3 WT and lane 4 *agp2Δ* were extracts prepared from cells treated with 3.5 μM of CHX for 15 minutes. An equal amount of protein was loaded on the gel and electrophoresed at 110 V until the dye front exited the gel. The analysis is representative of three independent experiments. Arrows indicated by the following colours red, purple, blue, green, and brown show the difference in protein levels between the WT and the *agp2Δ*.

variation in the expressed proteins was observed between the WT and *agp2Δ* strains irrespective of CHX treatment, although only a minimal variation could be observed within the WT treated (+) and WT untreated (-) subgroup or the *agp2Δ* treated (+) and *agp2Δ* untreated (-) subgroup of samples.

A detailed differential expression analysis was done to identify the significant differentially expressed proteins ($\log_2$fold $> = 2$ and alpha $< 0.05$) and 80 proteins with differential abundance (DA) were identified (S3 Table). The expression pattern of these proteins is represented in the form of a k-means ($k = 6$) clustered, protein-wise data-centered heatmap across all the sample replicates (Fig 3).

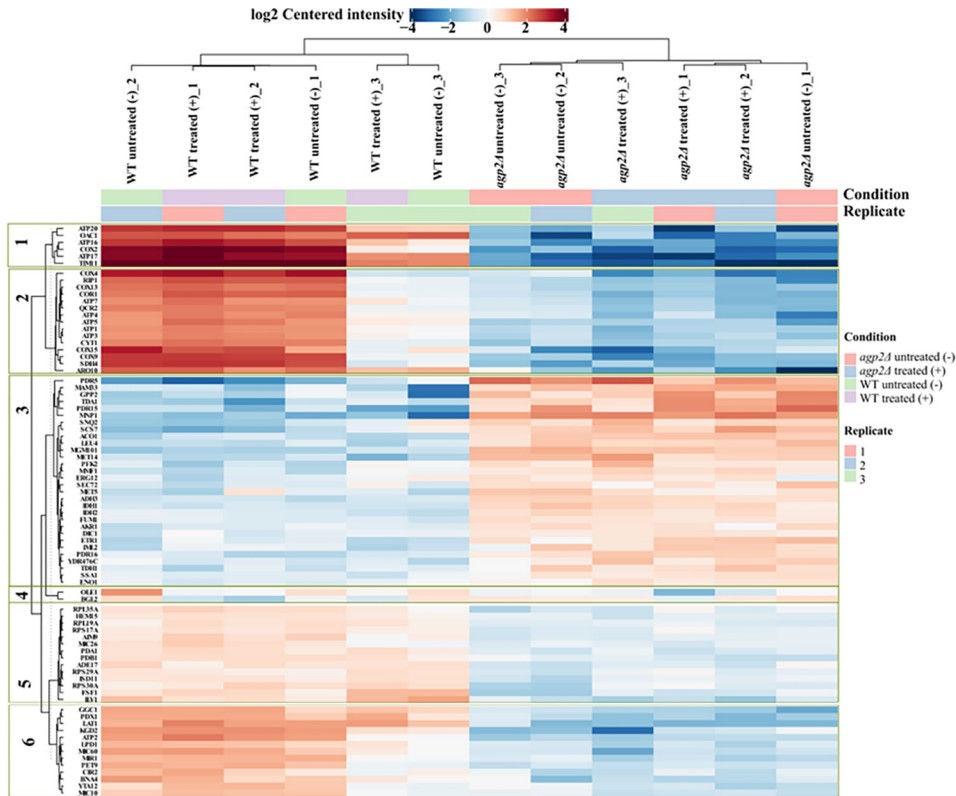

**Fig 3. Data-centered heatmap.** The significant differentially expressed proteins with log$_2$ fold change > = 2 and alpha < = 0.05 are visually represented as k-means clustered, protein-wise data-centered heatmap. Similar samples are clustered into column-based clusters while proteins with similar expression patterns are clustered into row-based clusters. The parameter $k$ = 6 was used and the proteins were enriched into 6 repertoires.

The 80 DA proteins were enriched into six protein groups based on the similitude of their expression patterns. Among the six protein clusters (S2 Fig, Clusters 1 to 6), four clusters (48 proteins in total), i.e., cluster 1 (6 proteins); cluster 2 (15 proteins); cluster 5 (14 proteins); and Cluster 6 (13 proteins) exhibited decreased expression, whereas, cluster 3 (30 proteins) exhibited increased expression in *agp2Δ* mutant compared to WT sample groups. Cluster 4 (2 proteins) exhibited mixed expression patterns in the WT and *agp2Δ* mutant sample groups.

### The *agp2Δ* mutant exhibits altered expression of diauxic shift proteins

Of the 80 DA proteins, 56 were mitochondrial proteins revealed by comparison with the known mitochondrial proteome dataset obtained from the three metabolic states of yeast cell growth (S3 Table) [41]. At least, 41 of the 56 mitochondrial proteins had decreased expression in the *agp2Δ* mutant sample group when compared to the WT sample groups. The other 15 proteins had increased expression in the *agp2Δ* mutant sample groups when compared to the WT sample groups. Furthermore, the 56 mitochondrial DA proteins were mapped to the list of proteins that are upregulated during the diauxic phase as yeast cells transition from fermentative (glucose phase) to respiratory (ethanol phase) [41]. Thus, the 41 mitochondrial proteins that showed decreased expression and the 15 that exhibited increased expression in the *agp2Δ* mutant as compared to the parent (S4 Table) displayed characteristics of diauxic shift proteins, suggesting that Agp2 may regulate the diauxic phase transition.

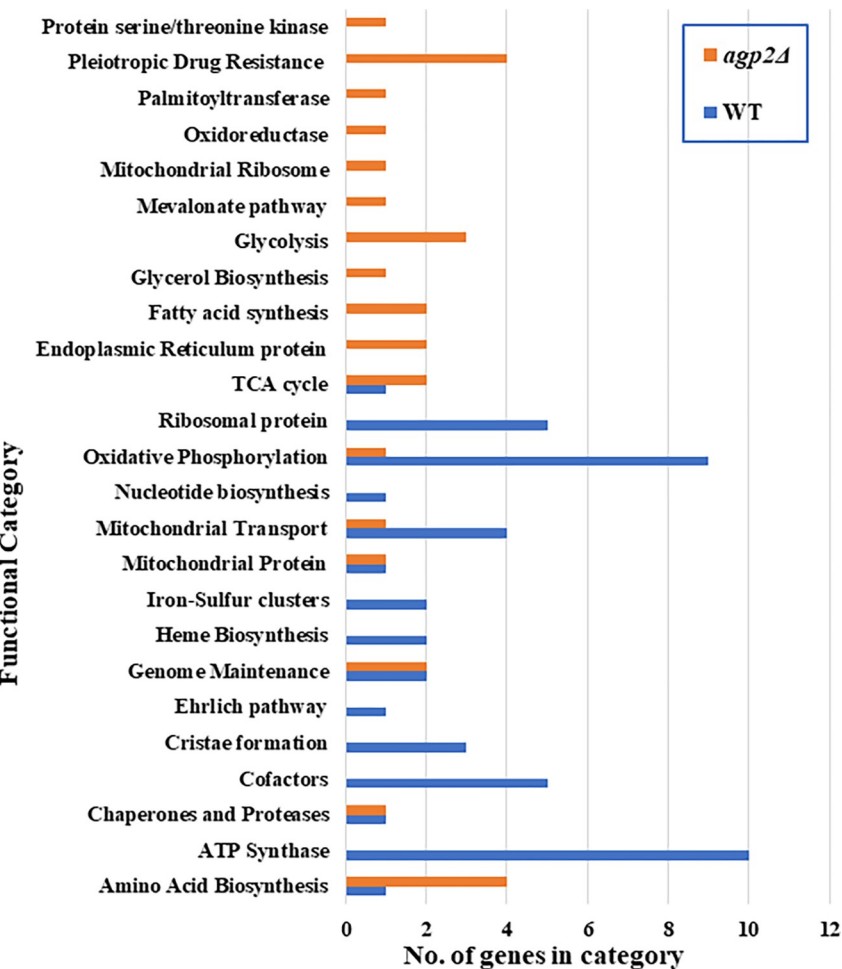

**Fig 4. Clustered bar chart of functional categories.** The 80 significant differentially expressed proteins were broadly categorized based on their functional annotations specified in the Saccharomyces Genome Database (SGD). The proteins highly expressed in WT and *agp2Δ* mutant are represented as blue and orange bars, respectively. The functional categories of these proteins represent the y-axis with the x-axis being the protein count in each category.

It is noteworthy that the metabolic transition into diauxic shift by yeast cells is characterized by certain indicative phenomena such as (i) activation of the tricarboxylic acid cycle and oxidative phosphorylation affected by glucose exhaustion, (ii) decreased expression of phosphofructokinase (Pfk2), (iii) up-regulation of ATP synthases and mitochondrial transporters, and (iv) up-regulation of gluconeogenesis and glyoxylate pathways induced by glucose depletion [42, 43]https://www.ncbi.nlm.nih.gov/pmc/articles/PMC4563728/. To further identify such phenomenon in the present dataset, the 80 DA proteins were broadly classified into different functional categories based on their annotated roles specified in the Saccharomyces Genome Database (SGD) (S5 Table). It can be observed that proteins classified under ATP synthase, oxidative phosphorylation, and mitochondrial transport were upregulated in WT sample groups indicating an efficient transition into the ethanol phase through the diauxic shift (Fig 4).

On the contrary, proteins involved in the glycolysis pathway such as Pfk2 and Eno1, which are characteristically downregulated in diauxic shift were upregulated in *agp2Δ* mutant sample groups (Fig 3 and S2 Fig, Cluster 3), supporting the notion that the *agp2Δ* mutant might be defective in the fermentation process.

## The *agp2Δ* mutant dysregulates the pleiotropic drug resistance pump Pdr5 and several mitochondrial proteins

We next conducted a multi-way comparison with the following six sample groups: (i) *agp2Δ* untreated (-) vs. *agp2Δ* treated (+); (ii) WT untreated (-) vs. WT treated (+); (iii) *agp2Δ* untreated (-) vs. WT untreated (-); (iv) *agp2Δ* untreated (-) vs. WT treated (+); (v) *agp2Δ* treated (+) vs. WT untreated (-); and (vi) *agp2Δ* treated (+) vs. WT treated (+). In this multi-way comparison, gene expressions were less contrasting between comparisons (i) and (ii), while 46 proteins were determined to be significantly different between the four comparisons (iii) to (vi) (at q<0.05) shaded in green (S1 Table). S3 Fig represents the expression pattern of proteins in the form of a k-means (*k* = 6) clustered, protein-wise data-centered heatmap across the four comparisons. Thirty-one proteins in Cluster 1 were upregulated, 6 proteins in Cluster 2 were highly downregulated, and 43 proteins in Cluster 3, Cluster 4, Cluster 5, and Cluster 6 were also downregulated in the four comparisons, i.e., (iii) to (vi). Several crucial proteins were found to be elevated in the *agp2Δ* mutant sample groups as compared to the WT sample groups as represented by the volcano plots (Fig 5 and S4–S6 Figs).

In the sample group comparison, *agp2Δ* untreated (-) vs. WT untreated (-), the differentially expressed proteins included the plasma membrane ATP binding cassette (ABC) transporters Pdr5 and its paralog Pdr15 involved in pleiotropic drug resistance, as well as the phosphatidylinositol transfer protein Pdr16 (Fig 5). Pdr5, Pdr15, and Pdr16 are all regulated by the multidrug resistance regulator Pdr1 (Fig 5) [44]. In addition, the levels of several mitochondrial proteins were altered in the *agp2Δ* mutant as compared to the WT. For example, the downregulated proteins in the *agp2Δ* mutant include a component of the pyruvate

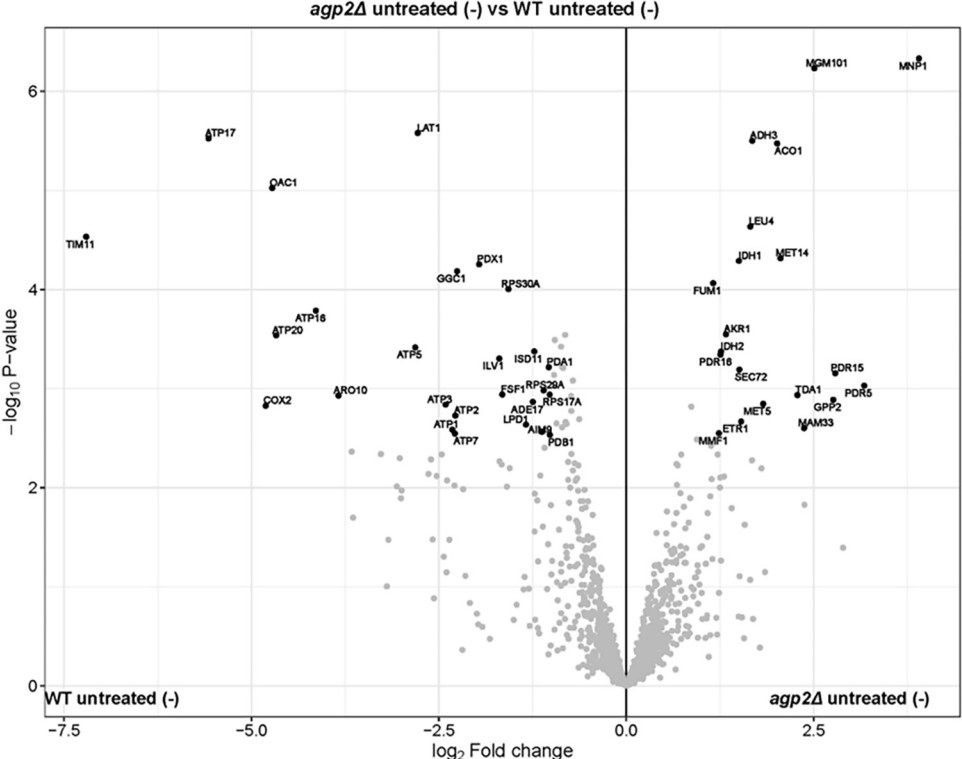

**Fig 5. Volcano-plot of *agp2Δ* untreated (-) vs. WT untreated (-).** The gene names of significant differentially expressed proteins with log$_2$ fold change > 0.5 and -log10 P-value > 2 are labeled.

dehydrogenase complex (Lat1), at least five subunits of the ATP synthase (subunits 5, alpha, beta, gamma, and delta, respectively ATP5, ATP1, ATP2, ATP3, and ATP7), the cytochrome c oxidase subunit 2 (Cox2), the mitochondrial dihydrolipoyl dehydrogenase Lpd1, and several more proteins, which are required for the proper functioning of the mitochondria (Fig 5).

The same pattern of altered protein levels was found in the *agp2Δ* mutant as compared to the WT when the analysis was done with the remaining three sample groups, i.e., the *agp2Δ* untreated (-) vs. WT treated (+); *agp2Δ* treated (+) vs. WT untreated (-); and *agp2Δ* treated (+) vs. WT treated (+) (see S4–S6 Figs). There were some minor exceptions, where a few unique proteins were represented in one sample group, but not another as in the case of the upregulated multidrug efflux pump Snq2 seen in the *agp2Δ* when comparing the *agp2Δ* treated (+) vs. WT treated (+) (S6 Fig). The protein Ole1 (Acyl-CoA desaturase 1), which influences cell wall permeability by fatty acid conversion, was significantly downregulated in *agp2Δ* treated (+) in contrast to WT untreated (-) (S5 Fig). Diminished Ole1 expression leads to decreased cell wall permeability due to increased fatty acid saturation. This is noted as an adaptive mechanism of yeast cells in response to herbicides [45]. Apart from these proteins, there was no significant difference when comparing either group with or without CHX. However, the major differences were between the *agp2Δ* mutant and the WT strain.

Collectively, the mass spectrometry data highlighted that (i) the membrane fractions contained proteins from the plasma membrane and the mitochondrial membrane, and (ii) the *agp2Δ* mutant caused a dysregulation leading to the elevated levels of the plasma membrane multidrug resistance pump Pdr5 and reducing the levels of critical mitochondrial proteins involved in energy production. Also, enrichment analysis (S7 Fig) of GO ontologies for bioprocesses in pairwise comparisons of the four sample groups revealed that the bioprocesses, which include transmembrane transport, ion transport, oxidative phosphorylation, ATP synthesis, and cellular respiration were the major contrasts between the *agp2Δ* mutant and the WT strain.

## RNAseq reveals differentially expressed genes in the *agp2Δ* mutant as compared to the WT

Since the mass spectrometry data revealed that the *agp2Δ* mutant displayed differences in the levels of several proteins as compared to the WT, we checked whether this could reflect changes in the gene expression pattern between these two strains. Total RNA was extracted from the *agp2Δ* mutant and the WT strain and subjected to bulk RNAseq analysis (see Materials and Methods). Due to the low read quality, one of the triplicate samples for the WT strain was excluded. A total of 188 million reads was obtained from the five samples with an average of 38 million reads in each sample (S6 Table). Nearly 5,744 genes (excluding snRNA and tRNA) with non-zero total read count across the samples (i.e. expressed in at least one sample) were observed in the RNAseq dataset and used for further analysis (S7 Table). The principal component analysis of the read counts of these genes revealed that there was a strong separation between the *agp2Δ* mutant and the WT strain, indicating that the two strains are different with respect to overall gene expression (S8 Fig). We used the DESeq2 Bioconductor package to perform a differential analysis of gene expression between the *agp2Δ* mutant and the WT. Several genes were differentially expressed between the strains as indicated in the MA plot (Bland-Altman plot for the visualization of genomic data) (S9 Fig) and represented by the volcano plot (Fig 6).

The analysis identified 322 differentially expressed genes (DEGs) that were significant with $log_2$ fold change $> = 2$ and adjusted *p*-value $< 0.05$ (S8 Table). Of these genes, 165 were upregulated and 157 were downregulated in *agp2Δ* mutant (S10 Fig). Thus, it appears that several physiological processes are likely defective in the *agp2Δ* mutant.

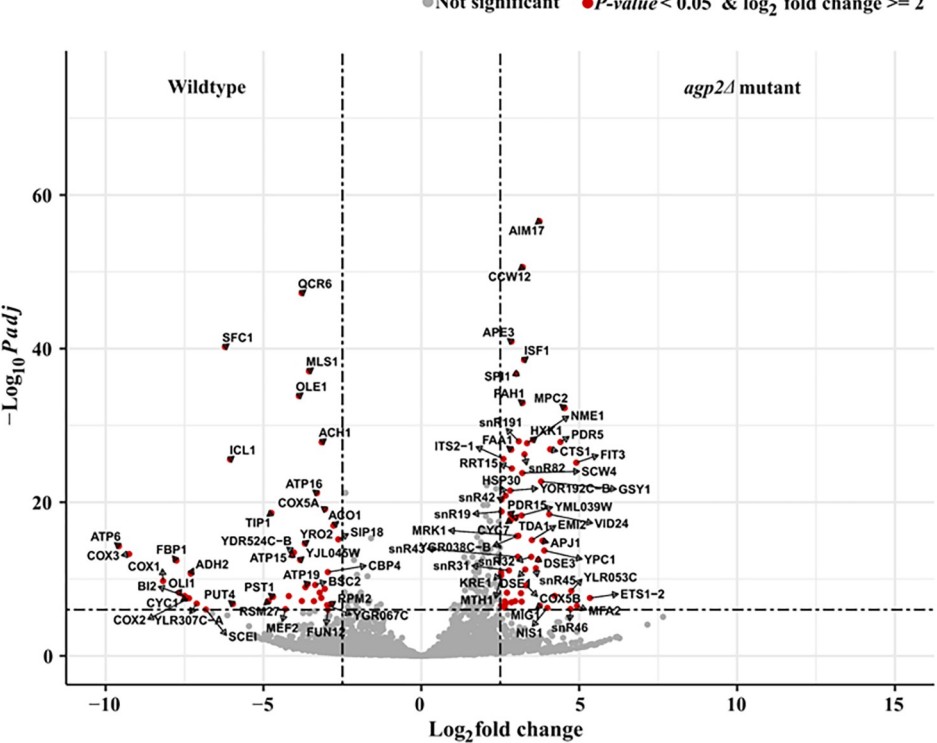

**Fig 6. Volcano plot.** Visual representation of the results of differential testing (*agp2Δ* mutant versus the WT) for gene expression (RNAseq derived gene dataset: 5744 genes). The data points are colored as described in the legend. The gene symbols of the data points upregulated/downregulated in *agp2Δ* mutant strain with log$_2$ fold change >2 and scoring top 15 P*adj* values are labeled.

## The *agp2Δ* mutant alters the regulation of several genes encoding transporter function

The significant DEGs were annotated into broad functional categories to identify the prominent categories that were upregulated/downregulated due to the deletion of the *AGP2* gene. Functional categorization of the genes was derived by mapping the gene symbols to the KEGG pathway ID in the org.Sc.sgd.db Bioconductor package (S8 Table) [46]. Genes with the missing KEGG pathway IDs were manually annotated by referring to the SGD website. The functional annotation "Transporters" for the DEGs with missing pathway IDs was derived by referring to the yeast transporter dataset of the TooT-SC bioinformatic tool that predicts the substrate class of a given transporter gene [47]. There were 50 significant transporter genes and the substrate class encoded by these genes is indicated in S8 Table. It is noteworthy that the pleiotropic drug transporter gene, *PDR5*, encoding the Pdr5 protein was among the top 60 DEGs elevated in the *agp2Δ* mutant (Fig 7), consistent with the mass spectrometry data (Figs 3 and 5), and hinting that it could be involved in the CHX resistance displayed by this mutant (see below).

The Uniform Manifold Approximation and Projection (UMAP) machine learning technique was used for dimension reduction and clustering of significant transporter genes with similar gene expression patterns. As shown in S11 Fig, the transporter genes were grouped into 5 clusters (2 clusters of upregulated genes and 3 clusters of downregulated genes in the mutant samples). Each cluster consisted of genes grouped irrespective of substrate classes. The pleiotropic drug transporter genes, *PDR5* and *PDR15* were part of a cluster of genes

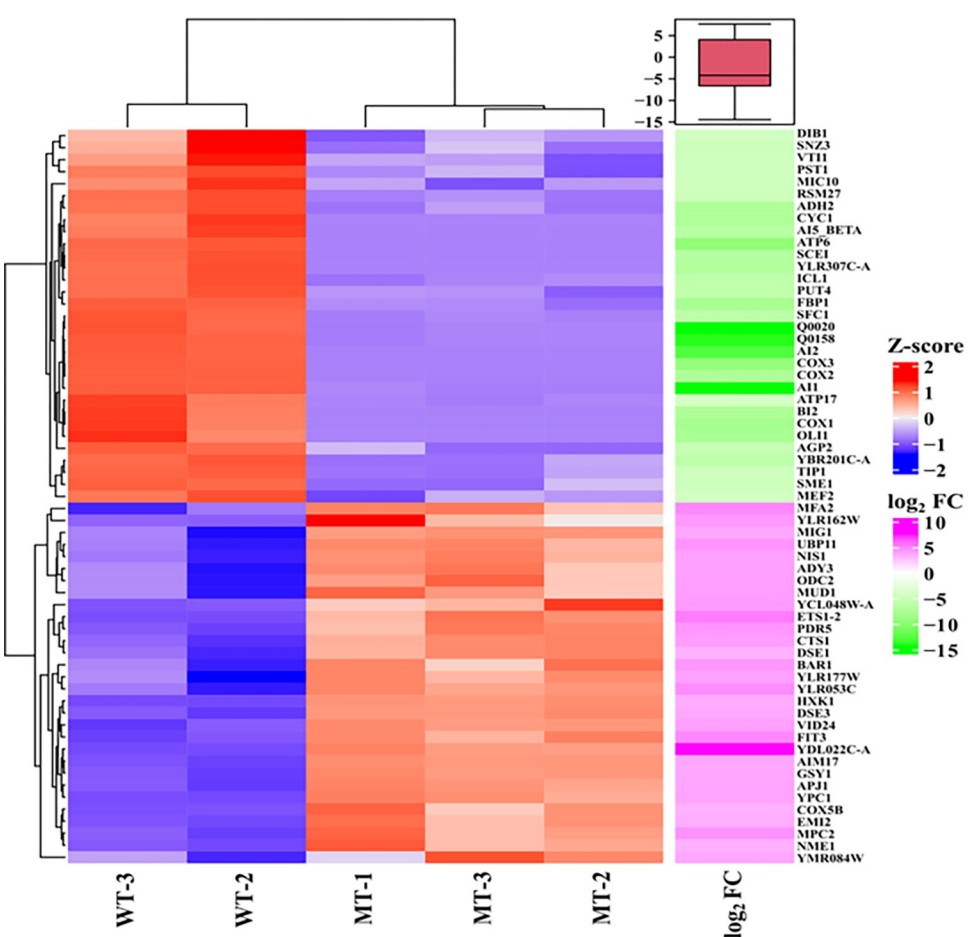

**Fig 7. Heatmap representing the expression levels of the top 60 significant genes along with their log₂ fold changes.**

upregulated in the *agp2Δ* mutant along with *STE6* (ABC protein), *COX5b* (mitochondrial electron transporter responsible for oxidative phosphorylation), *VMR1* (ABC vacuolar transporter), *HXT7* (glucose transporter), *MPC2* (mitochondrial pyruvate carrier, a crucial metabolic transporter), and HSP30 (a heat shock protein that also serves as a negative regulator of the proton pump Pma1). Thus, Agp2 can modulate the expression of several transporter genes.

### RT-PCR reveals that the *PDR5* gene is upregulated in the *agp2Δ* mutant

The RNAseq data indicated that the gene expression level of a key drug efflux pump Pdr5 documented to expel CHX from the cells was increased in the *agp2Δ* mutant (S8 Table). To confirm this, we used RT-PCR to monitor the *PDR5* gene expression level in the WT and the *agp2Δ* mutant. Briefly, total RNA was extracted from both strains, normalized to equal concentrations, and converted to cDNA. RT-PCR revealed that the *PDR5* gene was upregulated by 5-fold in *agp2Δ* strain, as compared to the wild-type, while the control gene *ACT1* was unchanged (Fig 8A and 8B).

While this finding confirms the mass spectrometry and RNAseq data, it raises the possibility of whether Agp2 could modulate *PDR5* gene expression by exerting control at the level of the *PDR5* promoter.

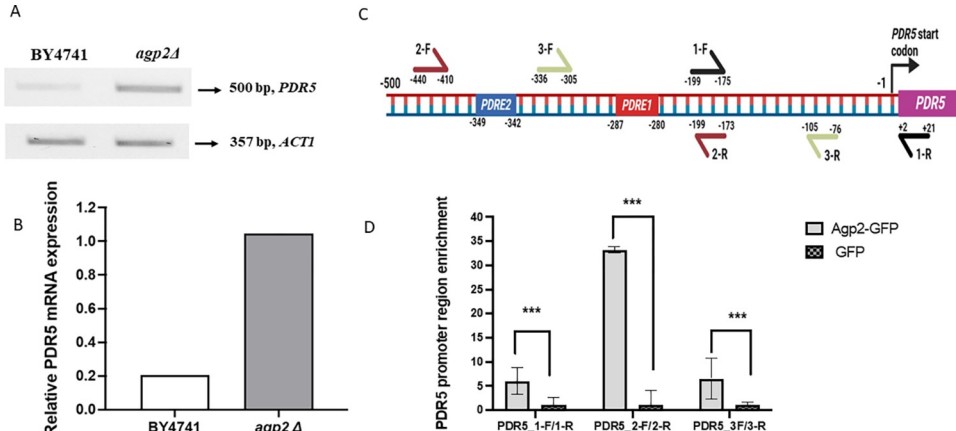

**Fig 8. *PDR5* gene expression is controlled by Agp2. A**, RT-PCR analysis showing the upregulation of the *PDR5* gene in the ***agp2Δ*** mutant. The ACT1 gene was used as a control. **B**, Quantification of the fold induction of the *PDR5* gene in the *agp2Δ* mutant. **C**, Promoter region of *PDR5* showing the location of the primer pairs for qRT-PCR used in the ChIP analysis. PDRE, Pleiotropic Drug Resistance Element PDRE1 5′–TCCGCGGA–3′ and the variant PDRE2 5′– TCCGTGGA–3′. **D**, Results of the ChIP analysis indicating the enrichment of Agp2 onto the 5'-upstream region, spanning– 440 bp to– 173 bp, of the *PDR5* gene. The analysis was repeated twice.

## Agp2 associates with the promoter of the *PDR5* gene using immunoprecipitation analysis

We checked whether Agp2-GFP could pulldown upstream sequences of the promoter region of the *PDR5* gene. To do this, we used the WT cells expressing either Agp2-GFP or carrying the vector expressing GFP alone and performed a ChIP analysis using an anti-GFP antibody. The pulldown material was probed for the upstream sequence of the *PDR5* gene using qRT-PCR and the primer sets shown in Fig 8C. This analysis revealed that Agp2-GFP was strongly associated with the region amplified with the primer pair 2-F and 2-R that spanned from—440 bp to—173 bp, while it showed less association with the primer pair 1-F and 1-R spanning from—199 bp to + 21 bp and the primer pair 3-F and 3-R spanning from—336 bp to —76 bp (Fig 8D). In contrast, the GFP protein showed no association with any of the primer sets. We interpret this finding to suggest that Agp2 is associated with the upstream region of the *PDR5* gene.

## The *agp2Δ* mutant downregulates several critical genes required for mitochondrial function

The RNAseq data also revealed that many genes involved in mitochondrial maintenance and energy production were downregulated in the *agp2Δ* mutant (S8 Table). These genes included *ATP6* and *ATP17* encoding subunits of the F1F0 ATP synthase; *COX1*, *COX2* and *COX3* encoding subunits of the cytochrome c oxidase complex of the mitochondrial electron transport chain involved in oxidative phosphorylation to produce ATP; *MIC10* encoding a mitochondrial protein of the intermembrane space that makes contact with the outer membrane; *RSM27* encoding a protein of the small subunit of the mitochondrial ribosome; and MEF2 encoding a mitochondrial translational elongation factor (Fig 7). In addition, 84 genes encoding mitochondrial proteins involved in the metabolic shift were differentially expressed in the *agp2Δ* mutant as compared to the parent. Of the 59 downregulated genes in the *agp2Δ* mutant, 46 are involved in diauxic shift and the ethanol phase (S9 Table) [43, 48, 49]. These data

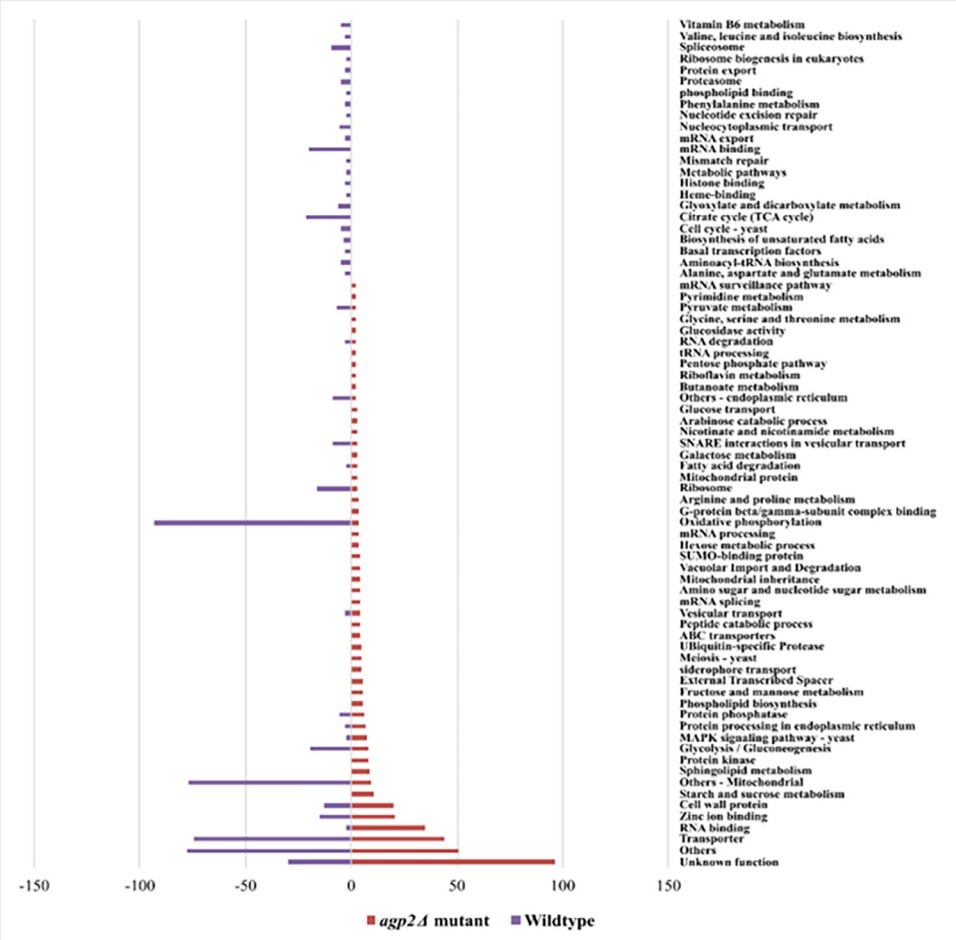

**Fig 9. Cumulative log₂ fold change profile of functional categories.** The functional categories of DEGs are represented on the y-axis and the cumulative log₂ fold change of all genes categorized into a particular functional category is shown on the x-axis.

suggest that mitochondrial functions would be compromised in the *agp2Δ* mutant and may explain its slow growth phenotype typical of petite mutants lacking mitochondrial functions.

To explore whether the *agp2Δ* mutant could be defective in mitochondrial function, we serially diluted overnight grown cultures of the parent and the *agp2Δ* mutant and spotted the strains onto solid media containing either glucose or the non-fermentable carbon source, glycerol (see S12 Fig). While the parent strain grew onto solid media containing either glucose or glycerol, in contrast, the *agp2Δ* mutant grew only onto solid media containing glucose and not the solid media containing glycerol as the carbon source (S12 Fig). This finding is consistent with the above analysis that the *agp2Δ* mutant is indeed defective in mitochondrial function.

It is noteworthy that the exploratory data analysis of both the count and cumulative log₂ fold change of the functional annotation categories revealed that most of the DEGs upregulated in the *agp2Δ* mutant belong to genes encoding proteins that (i) have unknown function, (ii) possess the ability to bind RNA, (iii) contain zinc as a co-factor, (iv) harbor protein kinase activity, and (v) display roles in the metabolism of starch, sucrose, mannose, and sphingolipid, for example, metabolic enzymes involved in gluconeogenesis (Hxk1, Glk1) and Hexose metabolic process (Emi2), and cell wall protein (Scw4) (S10 Table and Fig 9).

In contrast, the DEGs encoding other cellular and mitochondrial functions, aminoacyl-tRNA biosynthesis (Ths1 and YNL247W), nucleocytoplasmic transporter proteins (Tif3 and Fun12), ribosomal proteins and proteins involved in spliceosome and transporter proteins (Sec14, Ai1-2) were downregulated (S11 Table) [50–52]. Thus, it would appear that Agp2 plays a role in controlling the expression of genes belonging to diverse physiological pathways.

### The comparison between the RNAseq and mass spectrometry data identifies defects in common biological processes

The association between transcriptional and translational regulation of genes in yeast follows a dynamic pattern [53]. Hence a comparative data analysis was done to infer the differential expression of proteins deduced by mass spectrometry and the differential expression of genes deduced by RNAseq. From the mass spectrometry data analysis, 1122 protein-coding genes were identified and 80 of these genes were differentially abundant among the *agp2Δ* mutant and the WT sample groups. In contrast, RNAseq data analysis identified 5744 genes among which 322 genes were differentially expressed among the sample groups (S12 Table and S13 Fig). A comparison between the two analyses revealed that 4403 genes were unique to the dataset derived by RNA-sequencing and 23 genes were unique to the dataset derived by mass spectrometry. From the 80 protein-coding genes that were differentially abundant among the *agp2Δ* mutant and the WT sample groups, 75 were commonly identified by both the RNAseq and the mass spectrometry data analyses. From these 75 differentially expressed protein-coding genes, 21 were identified as significant in both analyses and 54 were identified as significant only in the RNAseq data analysis. It can be observed from S14 Fig that both the mRNA and protein expression of the Pdr5 efflux pump was upregulated in *agp2Δ* mutant when compared to WT.

We also further analyzed the GO ontologies for bioprocesses that were significantly enriched in the pairwise comparisons of *agp2Δ* mutant and the WT sample groups via RNAseq data analysis. Similar to mass spectrometry data analysis, many biological processes involved in transmembrane transport, aerobic respiration, ATP metabolic process, cellular respiration, carbohydrate metabolism, and mitochondrial electron transport were enriched among the differentially expressed genes (S13 Table and S7, S15 Figs). This is consistent with the notion that Agp2 is involved in regulating several biological processes and could be discerned by both the RNA levels derived by RNA sequencing and protein levels derived by mass spectrometry.

## Discussion

Recently, we provided *in silico* evidence that the protein synthesis inhibitor CHX can dock onto Agp2 with minimal energy requirement [23]. The association of CHX with Agp2 rapidly triggers its degradation, which has been monitored using the tagged form of Agp2, i.e., Agp2-GFP [23]. The CHX-induced degradation of Agp2-GFP occurred within a few minutes in a process marked by ubiquitinylation [23]. This is a common mechanism used by many other plasma membrane transporters and sensors such as the yeast Tat2, Gap1, Put4, and Mep2 that contain ubiquitinylated lysine residues with the branch modification Lys63 [10, 25, 54]. Agp2 could use a similar mechanism, as it has four ubiquitinylated residues and Lys595 harbors the branch Lys63 modification required for polyubiquitinylation and degradation [24, 26, 55, 56]. Thus, it would appear that turning off Agp2 function by proteolysis could be a critical process to reduce the entry of CHX into the cells and prevent further toxic effects of the drug. This possibility is consistent with the observation that cells lacking Agp2 are hyper-resistant to CHX (Fig 1) [5, 6, 23, 57]. This apparent mechanism does not preclude the possibility that Agp2 is acting (i) directly to transport CHX into the cells or (ii) indirectly via

activation of another uptake transporter(s) besides Dur3 and Sam3 [23] or (iii) via another pathway to control the uptake and efflux of CHX into and out of the cells, respectively. As such, we undertook the present study to investigate the possible mechanism by which the absence of Agp2 confers resistance to CHX.

Herein, we showed by mass spectrometric analysis that the total membrane fractions derived from the *agp2Δ* mutant contain altered levels of several proteins as compared to the parent strain (see volcano plot Fig 5). From our previously published study, we were expecting Agp2 to regulate an uptake transporter for CHX as it does for the uptake of polyamines [8]. Unexpectedly, we discovered that the pleiotropic drug efflux pump Pdr5, known to be involved in the extrusion of many toxic compounds from the cell, is a key candidate protein found to be upregulated in the *agp2Δ* mutant [58, 59]. We targeted Pdr5 as it was revealed by mass spectrometry showing the highest number of unique peptides (28 peptides) and confirmed by RNAseq. More importantly, it was shown to be involved in the efflux of CHX [58, 59]. Cells overexpressing Pdr5 are hyper-resistant to CHX, whereas, cells lacking Pdr5 are hypersensitive to the drug [60, 61]. As such, we expected that deleting the *PDR5* gene in the *agp2Δ* mutant would abolish the hyper-resistance of the *agp2Δ* mutant and the resulting *agp2Δ pdr5Δ* double mutant would be sensitive to CHX. However, we could not test this possibility as we were unable to delete the *PDR5* gene in the *agp2Δ* mutant to create the *agp2Δ pdr5Δ* double mutant strain. We believe that Pdr5 plays an essential function in the absence of Agp2, and so far, we have no explanation for this critical requirement of Pdr5. To date, we still do not know if Agp2 can directly transport CHX into the cells as labeled CHX is not available to test this possibility [6, 23]. However, it should be noted that the uptake transporters Dur3 and Sam3 are involved in controlling the sensitivity of the cells towards CHX, and hence, the entry of the drug into the cells [23]. The *agp2Δ* single mutant was substantially more resistant to CHX than the *dur3Δ sam3Δ* double mutant, suggesting that Agp2 could directly promote the uptake of CHX into the cells or via yet another uptake transporter, or activate the efflux of CHX.

To distinguish amongst the various aforementioned possibilities, we examined the global gene expression pattern between the *agp2Δ* mutant and the parent strain by RNAseq. The analysis revealed that the expression levels of 322 genes were affected including genes encoding four proteins Pdr5 (a multidrug resistant efflux pump), Pdr15 (a paralog of Pdr5), Pdr16 (a phosphatidylinositol transferase) and Snq2 (a plasma membrane ATP-binding cassette transporter involved in multidrug resistance) with documented roles in multidrug resistance. Pdr5, Pdr15, and Pdr16 are all regulated by the multidrug resistance regulator Pdr1. As such, Pdr5 was the primary focus due to its documented role in CHX resistance. The *PDR5* gene was conspicuously upregulated in the *agp2Δ* mutant consistent with its protein Pdr5 also being upregulated as detected by mass spectrometry. These two independent and complementary approaches unequivocally indicate that the upregulation of the Pdr5 protein is due to the increased expression of the *PDR5* gene in the *agp2Δ* mutant. RT-PCR confirmed that the *PDR5* gene expression was indeed upregulated, at least 5-fold, in the *agp2Δ* mutant. We interpret these data to suggest that under normal conditions, the Agp2 protein acts to repress the expression of the *PDR5* gene and upon exposure to CHX, Agp2 is rapidly degraded as we have previously observed [23] and this may relieve the *PDR5* repression (see the hypothetical models in Fig 10A and 10B).

This would promote the expression of Pdr5 to allow the cells to efflux CHX and establish cellular resistance to the drug (see the model in Fig 10). In such a mechanism, we postulated that a fraction of Agp2 would be localized in the nucleus and associated with the promoter of the *PDR5* gene (see the model in Fig 10A). We investigated this possibility by using chromatin-immunoprecipitation assay with an anti-GFP antibody and demonstrated that the immunoprecipitate from the *agp2Δ* mutant expressing Agp2-GFP, but not the vector control with

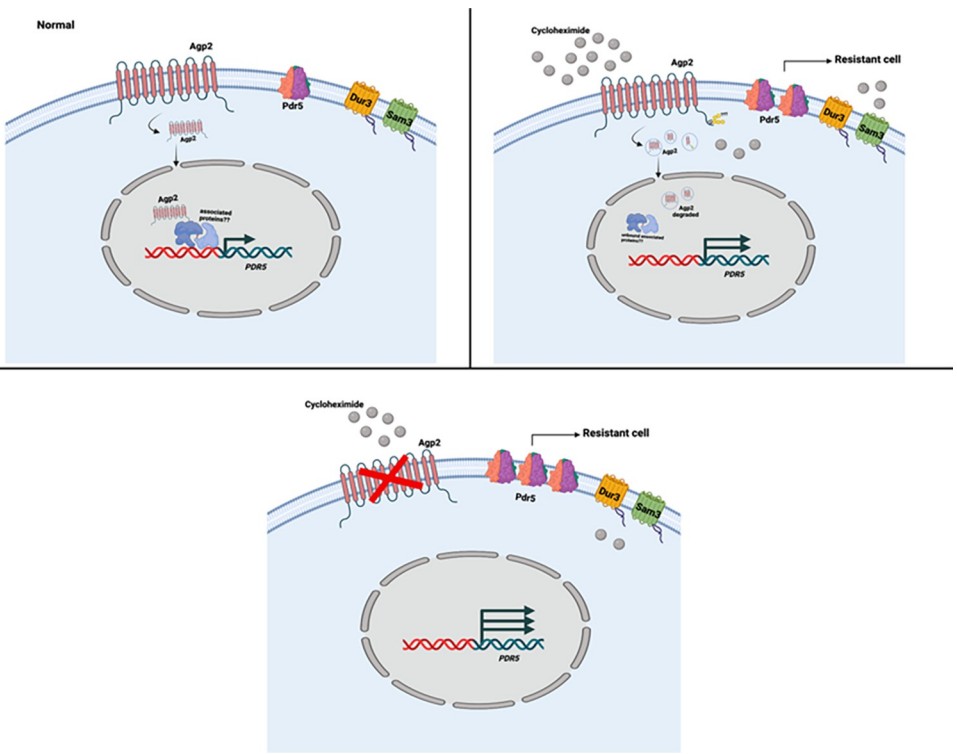

**Fig 10. The hypothetical models illustrate the unique roles of the permease Agp2 in response to stress caused by the fungicide CHX. A**, Agp2 is a plasma membrane-bound protein and a fraction is localized to the nucleus in association with the promoter of the *PDR5* gene. **B**, Agp2 can sense CHX, becomes rapidly ubiquitinylated, and triggers its degradation. This scenario would promote the expression of the *PDR5* gene. **C**, In the absence of Agp2, the PDR5 expression is depressed and the cells are resistant to CHX.

GFP alone, contained DNA regions of the *PDR5* promoter that were amplified by gene-specific primers. Agp2-GFP was enriched in the DNA segment spanning– 440 bp to– 173 bp of the *PDR5* promoter, but showed a weaker association with two other segments (Fig 8). We conclude that the hyper-resistance towards CHX displayed by the *agp2Δ* mutant is the result of the depression of *PDR5*, which is known to have a prominent role in the efflux of CHX (see the model in Fig 10). Thus, the hyper-resistance of the *agp2Δ* mutant towards CHX could be simply explained by the activated expression of the drug efflux pump Pdr5, and not necessarily that Agp2 possesses a direct role in CHX uptake into the cells.

Agp2 does not contain any apparent DNA binding domain and may suppress or activate gene expression by associating with transcriptional repressors and activators, respectively, as seen for the plasma membrane sensing complex Ssy1-Ptr3-Ssy5 that can trigger the activation of the transcription factors Stp1 and Stp2 [10, 13]. These transcription factors relocalized from the cytosol to the nucleus and cooperated with another transcription factor Dal82 to bind to the upstream activation sequence present in the promoters of the target genes such as *AGP1*, *BAP2*, and *TAT2* to trigger gene expression. Of relevance, the promoter region of the *PDR5* gene that is associated with Agp2 contains two pleiotropic drug resistance elements PDRE1 and PDRE2 bearing the classical sequence 5′–TCCGCGGA–3′ and the variant 5′ –TCCGTGGA–3′, respectively [62]. Both PDRE1 and PDRE2 can bind the paralogous zinc cluster transcriptional activators Pdr1 and Pdr3, which can form homodimers and or heterodimers to positively promote *PDR5* gene expression, as well as other ABC transporter genes such as *SNQ2* and *YOR1* [60–62]. Moreover, another zinc cluster protein Rdr1 is a transcriptional

repressor believed to bind to the same PDRE1 and PDRE2 sequenes as Pdr1 and Pdr3 to repress *PDR5* expression [63]. It is possible that under normal growth conditions, Rdr1 in association with Agp2 may prevent the constitutive binding of Pdr1 and Pdr3 to PDRE1 and PDRE2 thereby limiting the activation of the *PDR5* gene. Thus, in the absence of Agp2, Rdr1 may no longer retain the ability to bind to PDRE1 and PDRE2. However, we have not explored the possibility that the nuclear form of Agp2 is degraded in response to CHX, as we have shown recently for the plasma membrane form [23], in order to promote the expression of the *PDR5* gene.

Apart from the finding that cells devoid of Agp2 displayed upregulation of the *PDR5* gene, the mass spectrometry data revealed that of the 80 differentially expressed proteins 41 are associated with mitochondrial function and are downregulated in the *agp2Δ* mutant, which may explain the slow growth phenotype of the *agp2Δ* mutant. We considered this latter set as the next important group of proteins to be studied, as these are involved in a variety of mitochondrial functions that include transmembrane transport, ion transport, oxidative phosphorylation, ATP synthesis, and cellular respiration. This therefore explains the mitochondrial defect exhibited by the *agp2Δ* mutant, although we do not know the exact mechanism by which Agp2 maintains the expression of the mitochondrial genes. Herein, we showed that *agp2Δ* mutants are unable to grow on non-fermentable carbon sources such as glycerol consistent with its defective mitochondrial function (see S12 Fig). Since the mitochondria are the major source of energy supply for the cells, it seems plausible that several ATP-dependent processes are likely to operate at suboptimal levels. As such, the *agp2Δ* mutant could alter the regulation of alternate pathways to compensate for the deficiencies (Fig 9). It is noteworthy that *S. cerevisiae* $\rho^0$ cells with dysfunctional mitochondria lead to the elevated expression of the transcriptional activator gene *PDR3* and hence a strong upregulation of the targeted gene *PDR5* [64–66]. Based on the observation that the *agp2Δ* mutant is defective in mitochondrial function, the CHX resistance of this mutant might originate from the dysfunctional mitochondria leading to Pdr5 upregulation.

In short, the present data strongly support the notion that Agp2 possesses the ability to maintain gene expression for optimal growth. The exact mechanism by which Agp2 maintains regulatory control of genes, i.e., retaining expression of many genes while others are downregulated such as *PDR5*, is unclear. However, this would imply that Agp2 is associated with positive and negative regulators affecting a number of genes, and as such cells devoid of Agp2 are expected to exhibit resistance not only to CHX, but towards a wider range of drugs.

## Supporting information

**S1 Fig. PCA plot of top 500 significantly variable proteins.**
(PDF)

**S2 Fig. Bar plots of the differentially expressed proteins.**
(PDF)

**S3 Fig. Protein-wise data-centered heatmap for the contrasts between sample groups.**
(PDF)

**S4 Fig. Volcano-plot showing differentially expressed proteins comparing *agp2Δ* untreated (-) and wild type treated (+).**
(PDF)

**S5 Fig. Volcano-plot showing differentially expressed proteins comparing *agp2Δ* treated (+) and wild type untreated (-).**
(PDF)

**S6 Fig. Volcano-plot showing differentially expressed proteins comparing *agp2Δ* treated (+) and wild type treated (+).**
(PDF)

**S7 Fig. Bar plot of enriched GO ontologies for bioprocesses.**
(PDF)

**S8 Fig. Principal components analysis (PCA) plot of gene counts of 5744 protein-coding genes derived from RNAseq dataset.**
(PDF)

**S9 Fig. MA plot showing the log fold change of 5744 protein coding genes with respect to their mean gene expression between the *agp2Δ*mutant and wild type strains.**
(PDF)

**S10 Fig. Clustering of significantly expressed genes based on the Z-score of gene abundance analysis.**
(PDF)

**S11 Fig. UMAP (Uniform Manifold Approximation and Projection) plot for clustering of significant transporter genes.**
(PDF)

**S12 Fig. Spot test analysis showing that the *agp2Δ* mutant is unable to grow on solid YEPG media as compared to the WT strain BY4741.**
(PDF)

**S13 Fig. The Venn diagram represents the number of identified genes common among the mass-spectrometry and RNAseq-derived dataset.**
(PDF)

**S14 Fig. Heatmap of log$_2$ fold change derived from mass-spectrometry and RNAseq dataset.**
(PDF)

**S15 Fig. Bar plot of top 30 enriched GO ontologies for bioprocesses.**
(PDF)

**S1 Table. Mass-spectrometry results from Mascot.**
(XLSX)

**S2 Table. List of proteins with $\geq$5 unique peptide hits.**
(XLSX)

**S3 Table. Proteins that were significantly differentially expressed in *agp2Δ* in comparison to wild type irrespective of drug addition.**
(XLSX)

**S4 Table. Summary of the 57 differentially expressed mitochondrial proteins mapped to the list of upregulated proteins identified.**
(DOCX)

**S5 Table. Functional categories of 80 differentially expressed proteins derived by mass-spectrometry data analysis.**
(XLSX)

**S6 Table. Bulk RNA sequencing data statistics.**
(DOCX)

**S7 Table. Gene count matrix derived from RNAseq data processing.**
(XLSX)

**S8 Table. Genes upregulated and downregulated in the *agp2Δ* mutant strain.**
(XLSX)

**S9 Table. Total mitochondrial proteins affected by diauxic shift in the *agp2Δ* mutant vs. WT.**
(XLSX)

**S10 Table. Number of differentially expressed genes categorized in each functional category.**
(XLSX)

**S11 Table. Differentially expressed RNA binding proteins.**
(XLSX)

**S12 Table. Binary matrix representing the presence/absence of genes in the different gene lists derived from mass-spectrometry and RNAseq.**
(XLSX)

**S13 Table. Bioprocess Gene Ontologies (GO) of Differentially expressed genes identified by RNAseq data analysis.**
(XLSX)

**S1 Graphical abstract.**
(DOCX)

**S1 Raw images.**
(PDF)

## Acknowledgments

We thank the College of Health and Life Sciences, Hamad Bin Khalifa University, for providing scholarships to R.R., N.A., A.M., and A.A.S. Qatar National Library for defraying the article processing charge. The graphical model was created using Biorender.com.

## Author Contributions

**Conceptualization:** Dindial Ramotar.

**Data curation:** Yusra Manzoor, Mustapha Aouida, Ramya Ramadoss, Balasubramanian Moovarkumudalvan, Dindial Ramotar.

**Formal analysis:** Yusra Manzoor, Mustapha Aouida, Ramya Ramadoss, Balasubramanian Moovarkumudalvan, Nisar Ahmed, Abdallah Alhaj Sulaiman, Ashima Mohanty, Reem Ali, Borbala Mifsud, Dindial Ramotar.

**Funding acquisition:** Mustapha Aouida, Borbala Mifsud, Dindial Ramotar.

**Investigation:** Yusra Manzoor, Mustapha Aouida, Ramya Ramadoss, Balasubramanian Moovarkumudalvan, Abdallah Alhaj Sulaiman, Reem Ali, Dindial Ramotar.

**Methodology:** Mustapha Aouida, Ramya Ramadoss, Balasubramanian Moovarkumudalvan, Nisar Ahmed, Abdallah Alhaj Sulaiman, Ashima Mohanty, Reem Ali, Dindial Ramotar.

**Project administration:** Dindial Ramotar.

**Resources:** Ramya Ramadoss, Balasubramanian Moovarkumudalvan, Abdallah Alhaj Sulaiman, Dindial Ramotar.

**Software:** Ramya Ramadoss, Balasubramanian Moovarkumudalvan, Nisar Ahmed.

**Supervision:** Mustapha Aouida, Borbala Mifsud, Dindial Ramotar.

**Validation:** Yusra Manzoor, Mustapha Aouida, Ramya Ramadoss, Balasubramanian Moovarkumudalvan, Nisar Ahmed, Abdallah Alhaj Sulaiman, Ashima Mohanty, Reem Ali, Borbala Mifsud, Dindial Ramotar.

**Visualization:** Yusra Manzoor, Mustapha Aouida, Ramya Ramadoss, Balasubramanian Moovarkumudalvan, Nisar Ahmed, Abdallah Alhaj Sulaiman, Ashima Mohanty, Reem Ali, Borbala Mifsud, Dindial Ramotar.

**Writing – original draft:** Dindial Ramotar.

**Writing – review & editing:** Yusra Manzoor, Mustapha Aouida, Ramya Ramadoss, Balasubramanian Moovarkumudalvan, Nisar Ahmed, Abdallah Alhaj Sulaiman, Ashima Mohanty, Reem Ali, Borbala Mifsud, Dindial Ramotar.

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
