## [Decision Letter · Decision Letter 0]

17 Mar 2024

PONE-D-24-05515Loss of the yeast transporter Agp2 upregulates the pleiotropic drug-resistant pump Pdr5 and confers resistance to the protein synthesis inhibitor cycloheximide.PLOS ONE

Dear Dr. Ramotar,

Thank you for submitting your manuscript to PLOS ONE. After careful consideration, we feel that it has merit but does not fully meet PLOS ONE’s publication criteria as it currently stands. Therefore, we invite you to submit a revised version of the manuscript that addresses the points raised during the review process.

**ACADEMIC EDITOR: **

After considering the feedback from all the reviewers and my own assessment, I suggest proceeding with a major revision.

We look forward to receiving your revised manuscript.

Kind regards,

Vibhav Gautam

Academic Editor

PLOS ONE

Journal Requirements:

"Funds from Qatar Foundation provided to the College of Health and Life Sciences"

Additional Editor Comments:

After considering the feedback from all the reviewers and my own assessment, I suggest proceeding with a major revision.

Reviewers' comments:

Reviewer's Responses to Questions

**Comments to the Author**

1. Is the manuscript technically sound, and do the data support the conclusions?

Reviewer #1: Yes

Reviewer #2: No

Reviewer #3: Yes

2. Has the statistical analysis been performed appropriately and rigorously? 

Reviewer #1: I Don't Know

Reviewer #2: No

Reviewer #3: Yes

3. Have the authors made all data underlying the findings in their manuscript fully available?

Reviewer #1: Yes

Reviewer #2: No

Reviewer #3: Yes

4. Is the manuscript presented in an intelligible fashion and written in standard English?

Reviewer #1: Yes

Reviewer #2: No

Reviewer #3: Yes

5. Review Comments to the Author

Reviewer #1: The current research work investigates the possible mechanism by which the absence of Agp2 confers resistance to CHX. The work is comprehensively performed using multiple experimental and analytic approaches and is very well-written, introduced and discussed.

The following points need some consideration:

The graphical abstract should be improved to include the three sections as shown in Figure 10. It should be more representative and should clearly show that this is a hypothetical model.

Short title should be changed. It is different from the manuscript title and does not reflect the main topic being discussed in the article.

Nomenclature of genes and proteins should be carefully revised and corrected all over the manuscript.

Page 7: Conclusions of the study should not be reported in the introduction section.

There is detailed description of methodology in results section. Please make sure to remove excess details regarding methods from results and move that to methods section.

Immunoprecipitation results are mentioned in the discussion but not in the results section.

Page 33: The conclusion section should be more clear and specific.

Regarding figure 1B, the symbols on different growth curves and figure legend should be made clearer.

Page 15 Results section: In interpretation of figure 2, the authors state “suggesting that cells devoid of Agp2 may have defects in regulating protein expression levels.” Could you please clarify this sentence ? What do other arrows “the red, blue, green, and brown arrows” refer to ?

Among the upregulated proteins, why was Pdr5 specifically chosen, was this specifically targeted based on a pre-specified hypothesis to carry out the study? What are other genes of relevance from both mass spectrometry and RNA sequencing results ? Is there any overlap or shared genes ? Is there a priority or importance you find to any of these genes ?

Page 16: Authors mention “unique peptides and these showed a maximum fold difference of > 4.0” which you include in Supplementary Table S2. Do you find specific importance or physiologic relevance to any of these peptides? I believe this should be included in the discussion section.

Page 16: Authors also mention that “ four proteins are involved in the multidrug resistance pathway”. Why was Pdr5 specifically included and studied ?

Page 29 Discussion section: “ we showed by mass spectrometric analysis that the total membrane fractions derived from the agp2Δ mutant contain altered levels of several proteins as compared to the parent strain.” Could you please mention/discuss the most important of these proteins, its possible role, and its possible relation to the current point of study?

Page 28 : Again, authors states that “From these 75 differentially expressed protein-coding genes, 21 were identified as significant in both analyses” . On what bases was Pdr5 efflux pump chosen ?

Page 29: “Unexpectedly, we discovered that the pleiotropic drug efflux pump Pdr5, involved in the extrusion,……….” Why was this unexpected ?

Figure 10: I believe that this pathway of action is hypothetical and is a proposed mechanism or a possible interpretation for your findings. As it is not fully experimentally proved, you need to make it clear that this is a hypothesis in your figure title. “ A hypothetical Model”

Page 33: “and that cells devoid of Agp2 are expected to exhibit a wider range of phenotypes.” What is meant by this sentence ?

Reviewer #2: My comments on the manuscript entitled “Loss of the yeast transporter Agp2 upregulates the pleiotropic drug-resistant pump Pdr5 and confers resistance to the protein synthesis inhibitor cycloheximide” are as follows.

The study doesn't have clear proof showing that Agp2 is physically connected to the starting area of the PDR5 gene.

The difference in protein levels seen in membrane extracts from agp2Δ mutants might be affected by other things not related to Agp2. Without confirming these effects with other experiments, it's too early to say that the changes are only because Agp2 is missing.

The study doesn't consider other ways or backup plans that could also be causing the drug resistance seen in agp2Δ mutants. Without looking into these factors, the conclusion about Agp2's role in drug resistance might be too simple.

The idea that Agp2 breaking down leads to PDR5 going up and causing drug resistance needs more checking with tests that show how things work and with studies that fix the genetics.

The in silico evidence provided for the docking of CHX onto Agp2 and its subsequent degradation is based on computational simulations, which may not fully capture the complexities of protein-drug interactions in vivo. Experimental validation through biochemical assays or structural studies is necessary to confirm the binding and degradation kinetics observed.

While the degradation of Agp2-GFP in response to CHX treatment has been monitored, the exact mechanism of this degradation, including the involvement of ubiquitinylation, needs further experimental verification. Detailed biochemical assays are required to confirm the specific residues involved in ubiquitinylation and the role of Lys63 modification in Agp2 degradation.

The hypothesis that the degradation of Agp2 in response to CHX exposure reduces the entry of CHX into cells and prevents its toxic effects is plausible but requires direct experimental validation. Additional studies, such as cellular uptake assays with labeled CHX, are needed to confirm the role of Agp2 in CHX transport.

While RNAseq analysis revealed upregulation of the PDR5 gene in the agp2Δ mutant, the direct repression of PDR5 gene expression by Agp2 and its degradation in response to CHX exposure requires additional experimental validation. Further mechanistic studies, such as chromatin immunoprecipitation assays or reporter gene assays, are necessary to elucidate the regulatory role of Agp2 on PDR5 expression.

The authors are advised to revise and resubmit the manuscript following the recommendations outlined above.

Reviewer #3: This group previously reported that Agp2 mutant yeast are resistant to CHX. Agp2 mutation affects the uptake of CHX independently of Dur3 and Sam3. To further investigate the effect of Agp2 mutation, they produced a MS analysis of membrane bound proteins and found that the transporter Pdr5, which is involved in the efflux of CHX, is upregulated in the Agp2 mutant yeast. RNASeq results suggests that the PDR5 gene expression is increased in the Agp2 mutant yeast. They also present results suggesting that Agp2 could bind the promoter region of the PDR5 gene. The Agp2 mutation also affects mitochondrial function.

The manuscript is well written and informative. The results presented are providing new important information on the regulation of different transporters and their connections. The reader would benefit from a better support regarding the knowledge in this field of research. More information regarding the cellular localization of Agp2 and current knowledge on the regulation of Pdr5 protein and PDR5 gene, would improve the manuscript and the global comprehension. Many main figures are presenting general bioinformatic analyses performed on the results. Some data presented in supplementary figures or tables are important. They could be organized and presented in main figures. Fig 8 and text suggest that the PDR5 transcripts was quantified by RT-PCR but the description in methods suggests that it has been made by qRT-PCR. This should be clarified. In the model figure, should Agp2 be included in the nuclear membrane, or is it possible that the Agp2 transmembrane domain be disrupted or cleaved? Discussion regarding the possible reasons or mechanisms potentially involved in the Agp2 nuclear localization and binding to PRD5 promoter would help.

The discussion is interesting and very honest. It helps to understand the context and results presented.

6. PLOS authors have the option to publish the peer review history of their article (what does this mean?). If published, this will include your full peer review and any attached files.

Reviewer #1: No

Reviewer #2: **Yes: **Sachchida Nand Rai

Reviewer #3: No

---

## [Author Response · Author response to Decision Letter 0]

1 Apr 2024

We have also addressed all the specific requirements.

---

## [Decision Letter · Decision Letter 1]

16 Apr 2024

PONE-D-24-05515R1Loss of the yeast transporter Agp2 upregulates the pleiotropic drug-resistant pump Pdr5 and confers resistance to the protein synthesis inhibitor cycloheximide.PLOS ONE

Dear Dr. Ramotar,

Thank you for submitting your manuscript to PLOS ONE. After careful consideration, we feel that it has merit but does not fully meet PLOS ONE’s publication criteria as it currently stands. Therefore, we invite you to submit a revised version of the manuscript that addresses the points raised during the review process.

We look forward to receiving your revised manuscript.

Kind regards,

Vibhav Gautam

Academic Editor

PLOS ONE

Journal Requirements:

**Additional Editor Comments:**

Some minor comments still needs to be addressed.

Reviewers' comments:

Reviewer's Responses to Questions

**Comments to the Author**

1. If the authors have adequately addressed your comments raised in a previous round of review and you feel that this manuscript is now acceptable for publication, you may indicate that here to bypass the “Comments to the Author” section, enter your conflict of interest statement in the “Confidential to Editor” section, and submit your "Accept" recommendation.

Reviewer #1: (No Response)

Reviewer #2: (No Response)

Reviewer #3: All comments have been addressed

2. Is the manuscript technically sound, and do the data support the conclusions?

Reviewer #1: Yes

Reviewer #2: No

Reviewer #3: Yes

3. Has the statistical analysis been performed appropriately and rigorously? 

Reviewer #1: N/A

Reviewer #2: No

Reviewer #3: I Don't Know

4. Have the authors made all data underlying the findings in their manuscript fully available?

Reviewer #1: Yes

Reviewer #2: Yes

Reviewer #3: Yes

5. Is the manuscript presented in an intelligible fashion and written in standard English?

Reviewer #1: Yes

Reviewer #2: No

Reviewer #3: Yes

6. Review Comments to the Author

Reviewer #1: In Page 8 in tracked change copy and page 88 in the new PDF revised copy: Conclusions and results of the study has not been removed from the introduction section. Please remove the last paragraph of the introduction that show the results of the study.

Please include the response to comment 10 in the body of the manuscript.

Please ensure that the explanation shown in response to comment 13 is included in the discussion section as I cannot find it in tracked change copy.

Again, please make sure that genes and proteins names are correctly formatted allover the manuscript.

Reviewer #2: The revised manuscript is still not suitable for publication. In my original comments, i have already rejected the manuscript.

Reviewer #3: (No Response)

7. PLOS authors have the option to publish the peer review history of their article (what does this mean?). If published, this will include your full peer review and any attached files.

Reviewer #1: **Yes: **Wedad M. Nageeb

Reviewer #2: **Yes: **Sachchida Nand Rai

Reviewer #3: No

---

## [Author Response · Author response to Decision Letter 1]

24 Apr 2024

We have corrected two of the references.

---

## [Editor Report · Decision Letter 2]

1 May 2024

Loss of the yeast transporter Agp2 upregulates the pleiotropic drug-resistant pump Pdr5 and confers resistance to the protein synthesis inhibitor cycloheximide.

PONE-D-24-05515R2

Dear Dr. Ramotar,

We’re pleased to inform you that your manuscript has been judged scientifically suitable for publication and will be formally accepted for publication once it meets all outstanding technical requirements.

Kind regards,

Vibhav Gautam

Academic Editor

PLOS ONE

Additional Editor Comments (optional):

The manuscript is ready for publication since all comments have been addressed.
---

## [Editor Report · Acceptance letter]

10 May 2024

PONE-D-24-05515R2 

PLOS ONE

Dear Dr. Ramotar, 

I'm pleased to inform you that your manuscript has been deemed suitable for publication in PLOS ONE. Congratulations! Your manuscript is now being handed over to our production team.

Kind regards, 

on behalf of

Dr. Vibhav Gautam 

Academic Editor

PLOS ONE